# AutoWS-Bench-101: Benchmarking Automated Weak Supervision with 100 Labels

**Nicholas Roberts**[*], **Xintong Li**[*], **Tzu-Heng Huang, Dyah Adila, Spencer Schoenberg,**
**Cheng-Yu Liu, Lauren Pick, Haotian Ma, Aws Albarghouthi**[†]**, Frederic Sala**
University of Wisconsin–Madison
{nick11roberts, aws, fredsala}@cs.wisc.edu
{xli2224, thuang273, adila,
spencer.schoenberg, cliu547, lpick2, hma232}@wisc.edu

## Abstract

Weak supervision (WS) is a powerful method to build labeled datasets for training supervised models in the face of little-to-no labeled data. It replaces hand-labeling data with aggregating multiple noisy-but-cheap label estimates expressed by labeling functions (LFs). While it has been used successfully in many domains, weak supervision's application scope is limited by the difficulty of constructing labeling functions for domains with complex or high-dimensional features. To address this, a handful of methods have proposed automating the LF design process using a small set of ground truth labels. In this work, we introduce **AutoWS-Bench-101**: a framework for evaluating automated WS (AutoWS) techniques in challenging WS settings—a set of diverse application domains on which it has been previously difficult or impossible to apply traditional WS techniques. While AutoWS is a promising direction toward expanding the application-scope of WS, the emergence of powerful methods such as zero-shot foundation models reveals the need to understand how AutoWS techniques compare or cooperate with modern zero-shot or few-shot learners. This informs the central question of AutoWS-Bench-101: given an initial set of 100 labels for each task, we ask whether a practitioner should use an AutoWS method to generate additional labels or use some simpler baseline, such as zero-shot predictions from a foundation model or supervised learning.[3] We observe that in many settings, it is necessary for AutoWS methods to incorporate signal from foundation models if they are to outperform simple few-shot baselines, and AutoWS-Bench-101 promotes future research in this direction. We conclude with a thorough ablation study of AutoWS methods.

## 1 Introduction

The success of modern deep learning has been driven by large labeled datasets. Unfortunately, obtaining labeled data at scale is expensive and slow. Weak supervision (WS) [1, 2, 3, 4] is one of the most successful solutions to the problem of generating labeled datasets. The idea behind WS is simple: replace hand-labeling with acquiring and aggregating noisy-but-cheap label estimates from external sources—also called labeling functions (LFs)—to produce denoised estimates of true labels. WS is deployed widely and has had a significant impact. For example, in the medical domain, it has been used to re-create radiography datasets in a few hours that saved physician-months in

---

[*]Equal contribution

[†] Author's name in native alphabet: أوس البرغوثي

[3]The goal of generating the 101[st] label onward gives our benchmark its name.

36th Conference on Neural Information Processing Systems (NeurIPS 2022) Track on Datasets and Benchmarks.

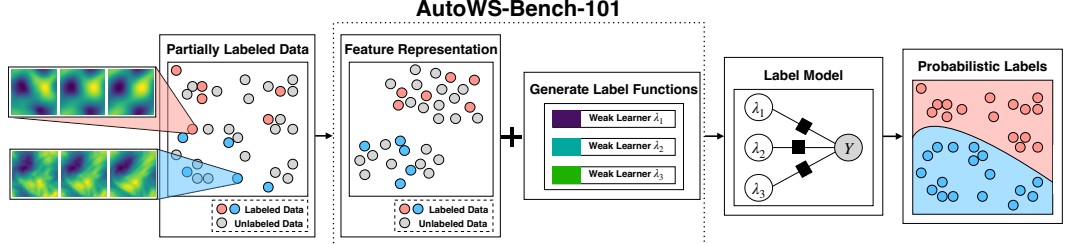

Figure 1: Overview of the AutoWS-Bench-101 pipeline.

manual labeling costs [5]. It has led to the creation of the largest dataset of aortic valve malformations developed atop the large-scale UK Biobank [6, 7]. In addition, WS has enjoyed wide industrial adoption. It is used in production by core teams within Google, Apple, and many more [8, 9].

Despite its massive success and widespread adoption, weak supervision can be challenging to apply. While much cheaper than manually labeling millions of examples, LF generation is sometimes costly. Obtaining high-quality LFs often requires training domain experts and painstaking iteration. Furthermore, creative choices must be made when applying WS to new modalities. For example, LFs that apply to text may not lift to image segmentation [10] or recommendation [11] tasks.

Automated WS (AutoWS) methods are a best-of-both-worlds solution. These techniques generate LFs automatically by training models on a small initial set of ground truth labels [12, 13, 14]. At first glance, AutoWS techniques resemble few-shot learners. However, they are much more, forming *omnivorous frameworks* capable of incorporating and combining signals from many sources, including powerful few- and zero-shot models. Indeed, recent evidence [15, 16] suggests that WS can exploit signal from powerful foundation models like CLIP, GPT-3, and T0++ [17, 18, 19].

AutoWS techniques have tremendous potential. For example, they

- can fuse multiple sources of signal, including foundation models, to maximize performance,
- inherit from WS the ability to denoise labels and to generalize beyond their constituent LFs,
- enable applying WS to domains and modalities where hand-crafting LFs is difficult or impossible.

Missing, however, is a thorough comparison of AutoWS techniques or an in-depth understanding of which components lead to good performance. For example, intuitively these methods rely on high-quality feature representations, but it is not clear how performance scales with the choice of representation. It is crucial to understand how these methods compare to—and can be combined with—existing few- or zero-shot models. Motivated to understand these mechanics under a diverse data settings previously underexplored by WS, we propose **AutoWS-Bench-101**.

**AutoWS-Bench-101** is a framework and benchmark for experimenting with, understanding, and evaluating AutoWS methods. It enables plug-and-play combinations of feature representations, LF candidate generation approaches, and LF selection procedures, conveniently built on top of the WRENCH [20] framework. It can be used with existing AutoWS methods like Snuba [12], interactive WS [13], and GOGGLES [14], or extensions of such techniques. A pipeline is shown in Figure 1. AutoWS-Bench-101 evaluation uses a diverse set of datasets on which WS has previously seen little application. Our benchmark reveals the following key insights about AutoWS:

1. Foundation models **are only helpful to AutoWS for in-distribution or related tasks,**
2. LFs that output multiple classes **might be better than class-specialized LFs,**
3. Foundation model usage can **hurt coverage,** the fraction of points labeled by AutoWS.

Once AutoWS methods are better understood, these methods have the promise of drastically increasing the application scope of WS. Our benchmark takes a crucial first step toward this understanding. We publicly release all experiment code and datasets for AutoWS-Bench-101 on GitHub.[4]

---

[4] https://github.com/Sala-Group/AutoWS-Bench-101

## 2 Background

We provide background on weak supervision (WS), automated WS (AutoWS), along with pre-trained and foundation models. Finally, we formalize the background on AutoWS by providing a mathematical formulation of the problem setting.

**Weak Supervision**   WS methods generate labels for unlabeled data by combining multiple sources of imperfect label estimates [1, 2, 4, 11]. These easily obtained and applied sources are called labeling functions (LFs). They can be created from heuristic rules produced by an expert, knowledge-base queries, pre-trained model outputs, or by any other means cheaper than obtaining ground-truth labels [21, 22, 23, 24, 25]. LFs provide a noisy estimate of the label, or otherwise abstain. If all the LFs abstain on a point, WS methods cannot label it; the fraction of points labeled by WS is the *coverage*.

Much of the WS literature focuses on the label model, the tool used to aggregate LF outputs to produce de-noised label estimates. New variants with different properties were introduced in [1, 3, 4, 11]. The recent benchmark WRENCH [20] focuses on evaluating such models. In contrast, we are focused on understanding the mechanics of AutoWS independent of the choice of label model.

**Automated Weak Supervision**   In many cases, the process of designing LFs is expensive or challenging. Several automated weak supervision (AutoWS) methods have been developed to automatically create LFs from some small initial set of labeled examples. Snuba [12] generates a large set of weak learners trained on combinations of features and iteratively selects LFs based on various performance criteria. Interactive Weak Supervision (IWS) [13] also selects LFs from a pool of candidates by interactively learning from human feedback. While not explicitly an automated procedure, IWS is easily modified to not require a human in the loop. Finally, GOGGLES [14] was introduced to address the challenge of automating the LF creation process, as well as the challenge of using WS for image data. GOGGLES computes pairwise affinity scores between feature representations of examples obtained by, for example, a pretrained VGG16 network, and learns a hierarchical generative model on the resulting affinity matrices. Additionally, several AutoWS methods have been developed specifically for the NLP domain. Some methods such as Darwin [26], PRBoost [27], and Nemo [28], like IWS, query annotators for feedback to assess the quality of synthesized rules. Other frameworks can be more complex and even more tailored to the NLP domain, such as ARI, which integrates inferred rules into pretrained NLP models [29].

**Pre-trained and Foundation Models**   Powerful pre-trained models are increasingly useful in modern machine learning. A standard approach to sidestep acquiring huge labeled datasets is to fine-tune a pre-trained model. For image data, this might be a ResNet [30] trained on ImageNet. Self-supervised models [31, 32] can be used to acquire high-quality representations enabling users to train a simpler downstream model using fewer labeled points. Foundation models [33], large-scale models trained on vast amounts of data, can be used as few- or zero-shot predictors. These approaches can be thought of as either competitors of AutoWS—if used as alternatives—or as AutoWS *ingredients*. For example, WS can be performed on top of foundation model embeddings, or use them to extend LF coverage [15], or use prompts as sources [16]. Understanding how pre-trained or foundation models compete with or contribute to AutoWS is a key element of this work.

**Problem formulation**   We provide a mathematical formulation of the AutoWS problem setting here. First, we begin with an unlabeled training set of $n$ examples $X_{\text{train}} = [x_i]_{i=1}^n$ with $x_i \in \mathcal{X}$ and a labeled validation set of $m$ examples $X_{\text{val}} = [x_j]_{j=n+1}^{n+m}$ with $x_j \in \mathcal{X}$, $Y_{\text{val}} = [y_j]_{j=n+1}^{n+m}$ with $y_j \in \mathcal{Y}$. Next, we embed examples in $\mathcal{X}$ into $\mathbb{R}^d$ using a feature extractor, which we denote as $\phi : \mathcal{X} \to \mathbb{R}^d$. Additionally, we denote the hypothesis class of synthetic LFs as $\mathcal{H}$ where $\forall h \in \mathcal{H}$, we have $h : \mathbb{R}^d \to \mathcal{Y} \cup \{-1\}$. We will denote an AutoWS method as a pair of algorithms, the first of which generates LFs and is parameterized by a feature extractor: $\mathcal{A}_{\text{WS}}^\phi = \left( \text{LF}_{\text{WS}}^\phi, \text{LM}_{\text{WS}} \right)$ where $\text{LF}_{\text{WS}}^\phi : \mathcal{X}^{n+m} \times \mathcal{Y}^m \to \mathcal{P}(\mathcal{H})$ and the second is the label model, which infers a weak label from one or more LF predictions: $\text{LM}_{\text{WS}} : \mathcal{Y}^{\geq 1} \to \mathcal{Y}$. We generate LFs using $\Lambda = \text{LF}_{\text{WS}}^\phi(X_{\text{train}}, X_{\text{val}}, Y_{\text{val}})$ and obtain LF predictions $\forall h_k \in \Lambda$ as $\lambda_i^k = h_k(\phi(x_i))$. We can furthermore obtain additional label predictions for each example from other sources, such as zero-shot predictions from foundation models or by using hand-designed LFs. We will denote the set of these additional predictions as

$\widetilde{\lambda}_i = \left\{ \widetilde{\lambda}_i^l \right\}_{l \geq 0}$, $\forall i \in [n + m]$. Finally, we obtain weak labels for the training set using the label model: $\widehat{y}_i = \mathrm{LM}_{\mathrm{WS}}(\{\lambda_i^k\}_{k=1}^{|\Lambda|} \cup \widetilde{\lambda}_i)$, $\forall i \in [n]$. In this work, we evaluate choices of $\phi$ and $\mathcal{A}_{\mathrm{WS}}^{\phi}$.

## 3 Benchmark Goals and Methodology

We describe the goals of our proposed benchmark and our experimental methodology.

**Goals**  The goal of this work is to understand the effects of the key components of AutoWS—LFs, selection methods, and representations—on a diverse collection of datasets. We are especially interested in settings far from the ambit of conventional weak supervision—most often used in text applications. We also seek to understand how AutoWS compares to leading zero-/few-shot methods when used as either an alternative or an ingredient.

**Methodology**  For ease-of-use, our benchmark is implemented as an extension of the recently introduced Weak supeRvision bENCHmark (WRENCH) [20]. We note that WRENCH and AutoWS-Bench-101 have fundamentally different goals: WRENCH evaluates label models, not AutoWS techniques. An additional difference is that WRENCH uses fixed pre-computed LFs and only includes a handful of image-based datasets without access to the original source images or videos.

We consider three well-known methods for generating LFs: Snuba, IWS, and GOGGLES [12, 13, 14], operating on a handful of baseline features: raw pixels, PCA with 100 principal components, the logits layer of a pre-trained ResNet-18 [30], and zero-shot logits from CLIP [17]. The AutoWS-Bench-101 pipeline is shown in Figure 1. Below, we explain our choices in more detail.

### 3.1 Datasets

In order to evaluate AutoWS techniques on a range of challenging WS settings, we include a diverse set of 10 classification datasets. Image classification datasets are a salient example of the types of datasets that fit our goal, as these have proven to be a challenging domain for weak supervision. We include two such datasets, MNIST and CIFAR-10 [34, 35], in AutoWS-Bench-101. We wish to avoid AutoWS methods overfitting to the image domain through the exploitation of feature representations specifically tailored to these tasks. For this reason we include two image-derived datasets that move beyond the scope of natural image classification. Spherical MNIST [36] consists of MNSIT digits stereographically projected onto the sphere, rotated, and mapped back to the plane. Permuted MNIST consists of MNIST digits whose pixels are permuted by a fixed bit-reversal permutation applied to the rows and columns. We additionally include three text datasets that are also used in WRENCH [20]: YouTube [37], Yelp [38, 39], and IMDb [40, 39]. Finally, we include three tasks in more diverse application settings beyond image, image derived, or text datasets. These are the electrocardiogram (ECG) dataset of [41], a turbulence classification variant of the Navier-Stokes PDE data used in [42], and EMBER, a tabular malware detection dataset [43]. Further details of the latter three datasets are provided below.

**ECG**  The ECG, or electrocardiogram dataset is commonly used to detect heart disease. It contains 60 second or shorter ECG recordings posed as a four-class classification problem: normal, disease, other, or noisy. This dataset was derived from the 2017 PhysioNet Challenge [41]. We use a fixed size of sliding window to process signals into a 1,000 ms time-series dataset.

**Navier-Stokes**  The Navier-Stokes equation is a partial differential equation that describes viscous fluid flows. A property of interest to the computational fluid dynamics community is the turbulence of these flows. In the case of the Navier-Stokes equation, this is governed by a viscosity parameter $\eta$. Navier-Stokes solutions are generated by [42]. We adapt the Navier-Stokes dataset at $\eta = 1\mathrm{e}{-3}$ (non-turbulent) and $\eta = 1\mathrm{e}{-5}$ (turbulent) to be a binary turbulence classification task.

**EMBER**  The EMBER dataset, or Elastic Malware Benchmark for Empowering Researchers, introduced by [43], is a tabular malware detection dataset extracted from malicious and benign Windows portable executable files. EMBER dataset is a binary classification task, and it includes

several groups of feature, such as parsed features, format-agnostic features, and string information to detect malware. We use the open source implementation to extract features.[5]

## 3.2 Implementation Details

We adapt the open source implementations of Snuba, IWS, and GOGGLES to be compatible with WRENCH [12, 13, 14, 20]. As AutoWS methods require a small set of labeled examples, we use 100 labeled examples in all experiments unless otherwise stated. We include a few-shot learning baseline—a logistic regression classifier trained on the same 100 labeled examples. Additionally, we include a semi-supervised baseline—label propagation [44]. Finally, we include zero-shot CLIP as another baseline. In the final evaluation of each method, we report output label model accuracy (see Table 1) and coverage (see Appendix).

**Snuba and Multiclass Snuba** Unless otherwise stated, we use the Snuba implementation and hyperparameters [12]. This implementation only supports binary classification—however, five our datasets are multiclass. We adapt Snuba to multiclass settings in two ways. First, we run Snuba's LF synthesis procedure for each class as a one-vs-all problem. This results in **unipolar** LFs—a given LF will either vote for its assigned class or abstain. We also implemented a **multipolar** version of Snuba where LFs are themselves multiclass. By default, the search space of LFs is fixed to be decision stumps, logistic regression, or nearest neighbor classifiers. We found that allowing Snuba to consider both decision stumps and logistic regression classifiers for any given LF led to a slight improvement in performance. All of our experiments consider both of these. Each LF abstains according to a confidence threshold that is inferred by Snuba during the LF synthesis procedure. Finally, as in the original Snuba implementation, we aggregate the outputs of each LF using the label model.

**Interactive Weak Supervision** We adapt the original implementation of IWS to our setting. IWS operates on an initial pool of binary LFs. For simplicity, we use Snuba's LF synthesis procedure to generate a large set of LFs that IWS can select from. Note that since we use Snuba's LF synthesis procedure, each LF has its own inferred abstention threshold. Specifically, we generate LFs with cardinality $D = 1$, the number of features that each LF can consider. Similarly, unless otherwise stated, we consider both decision stumps and logistic regression classifiers. Like Snuba, the original IWS implementation only supports binary classification. Since our benchmark is over 10-class problems, we run the IWS LF selection procedure once per class to produce unipolar LFs for each class before passing the outputs of these to the label model.

**GOGGLES** GOGGLES computes pairwise affinity scores on a set of unlabeled examples which are used in a hierarchical generative model with to infer labels. To construct the affinity matrix, the default implementation uses the max-pooling layers of the pre-trained VGG-16 network to extract various levels of semantic representations. In our experiments, we replace these VGG-16 representations with our baseline feature representations. Pairwise affinity scores are computed by cosine similarity between embeddings of each instance. After training a generative model to create clusters, GOGGLES uses a small amount of labeled datapoints to map clusters to classes. Note that GOGGLES does not abstain.

**CLIP** We used standard CLIP [17] zero-shot prediction as a baseline, comparing image embeddings (except for the tabular dataset EMBER) with text embeddings of strings related to the class label. The complete list of these strings is provided in the Appendix.

**BERT embeddings for text datasets** Our pipeline requires the features of each dataset to be compatible with all (or most) of the feature representation techniques that we evaluate. In order to make the text datasets that we evaluate work with our pipeline, we first extract feature vectors for our text datasets using BERT before applying other feature representations (except in the case of CLIP, which operates on text directly). For YouTube, Yelp, and IMDb, raw features refers to these BERT features, PCA embeddings refers to the PCA of BERT features, and ResNet-18 embeddings refers to BERT features passed through a pretrained ResNet-18 model. In the case of CLIP, we directly use the CLIP text encoder for both the input text and the candidate labels to produce a logits vector.

---

[5]`https://github.com/elastic/ember`

Table 1: Test accuracies of AutoWS methods across features and datasets using 100 labels. N/A denotes an incompatibility between a method, feature representation, and/or task—see Appendix. **Bolded** and underlined scores indicate the **first** and second best results, respectively.

| | MNIST | CIFAR-10 | SPHERICAL MNIST | PERMUTED MNIST | ECG | NAVIER-STOKES | EMBER | YOUTUBE | YELP | IMDB |
|---|---|---|---|---|---|---|---|---|---|---|
| **GOGGLES** | | | | | | | | | | |
| +RAW FEATURES | 0.3778 | 0.1697 | 0.0897 | 0.3778 | 0.2710 | 0.5284 | 0.5283 | 0.5880 | 0.4732 | 0.5516 |
| +PCA | 0.4563 | 0.1989 | 0.1074 | 0.4561 | 0.2688 | 0.4863 | 0.6728 | 0.5880 | 0.5466 | 0.5628 |
| +RESNET-18 | 0.3258 | 0.1768 | 0.1192 | 0.1894 | 0.3127 | 0.9353 | 0.5312 | 0.5200 | 0.4553 | 0.5152 |
| +CLIP | 0.3309 | 0.5436 | 0.1884 | 0.1405 | N/A | 0.7505 | N/A | 0.5280 | 0.4779 | 0.5000 |
| **IWS** | | | | | | | | | | |
| +RAW FEATURES | 0.4128 | 0.1482 | 0.1154 | 0.3862 | 0.2615 | 0.5403 | 0.4997 | 0.4896 | 0.6034 | 0.5478 |
| +PCA | 0.3503 | 0.1047 | 0.1283 | 0.4970 | 0.2687 | 0.5155 | 0.4777 | 0.5217 | 0.5260 | 0.5045 |
| +RESNET-18 | 0.0822 | 0.2120 | 0.1352 | 0.5101 | 0.4235 | 0.7995 | 0.7343 | 0.7840 | 0.5620 | 0.5286 |
| +CLIP | 0.6885 | **0.7892** | 0.1108 | 0.3495 | N/A | N/A | N/A | N/A | N/A | N/A |
| **SNUBA** | | | | | | | | | | |
| +RAW FEATURES | 0.2694 | 0.1624 | 0.0805 | 0.2308 | 0.2648 | 0.5235 | 0.7082 | 0.7520 | 0.6034 | 0.5936 |
| +PCA | 0.2313 | 0.0924 | 0.1526 | 0.2640 | 0.2598 | 0.8811 | 0.4905 | 0.6640 | 0.7187 | 0.5560 |
| +RESNET-18 | 0.4219 | 0.1520 | 0.1554 | 0.1804 | 0.2506 | 0.9774 | 0.6994 | 0.7400 | 0.5674 | 0.5804 |
| +CLIP | 0.6617 | 0.7550 | **0.3103** | 0.2419 | N/A | 0.8414 | N/A | 0.6280 | 0.5608 | 0.5699 |
| **SNUBA (MULTIPOLAR)** | | | | | | | | | | |
| +RAW FEATURES | 0.2374 | 0.1386 | 0.0991 | 0.2343 | 0.2596 | 0.5421 | 0.7264 | 0.7080 | 0.6484 | 0.6072 |
| +PCA | 0.5625 | 0.1390 | 0.2354 | 0.5625 | 0.2644 | 0.8968 | 0.5143 | 0.6280 | 0.7218 | 0.5020 |
| +RESNET-18 | 0.3866 | 0.2036 | 0.1588 | 0.2283 | 0.3702 | 0.9742 | **0.7559** | 0.7320 | 0.5526 | 0.5864 |
| +CLIP | 0.3541 | 0.6788 | 0.2479 | 0.2785 | N/A | 0.7916 | N/A | 0.7759 | 0.5435 | 0.5681 |
| **FEW-SHOT (LOGISTIC)** | | | | | | | | | | |
| +RAW FEATURES | 0.7295 | 0.2315 | 0.1496 | 0.7295 | 0.2583 | 0.6316 | 0.6517 | **0.8920** | **0.7871** | **0.6472** |
| +PCA | 0.7306 | 0.2262 | 0.1483 | **0.7305** | 0.2548 | 0.7863 | 0.7488 | 0.8880 | 0.7868 | 0.6464 |
| +RESNET-18 | **0.7966** | 0.3335 | 0.2544 | 0.3485 | **0.4700** | **1.0000** | 0.5329 | 0.7000 | 0.5682 | 0.5856 |
| +CLIP | 0.3579 | 0.6857 | 0.1011 | 0.1009 | N/A | 0.7658 | N/A | 0.7000 | 0.6347 | 0.5636 |
| **SEMI-SUPERVISED (LABEL PROP.)** | | | | | | | | | | |
| +RAW FEATURES | 0.7066 | 0.1927 | 0.1374 | 0.7066 | 0.2644 | 0.5010 | 0.5003 | 0.8440 | 0.5447 | 0.5812 |
| +PCA | 0.7165 | 0.1914 | 0.1405 | 0.7156 | 0.2644 | 0.5011 | 0.5003 | 0.8440 | 0.5450 | 0.5812 |
| +RESNET-18 | 0.7124 | 0.1277 | 0.1283 | 0.2106 | 0.2644 | 0.9700 | 0.5003 | 0.7000 | 0.5224 | 0.5500 |
| +CLIP | 0.1178 | 0.5780 | 0.1009 | 0.1009 | N/A | 0.5058 | N/A | 0.6480 | 0.5737 | 0.5400 |
| **CLIP ZERO-SHOT** | 0.5817 | 0.6989 | 0.2065 | 0.0899 | N/A | 0.6305 | N/A | 0.472 | 0.5223 | 0.5008 |

## 4 Analysis

AutoWS-Bench-101 evaluates combinations of AutoWS methods and feature representation methods on a diverse set of classification tasks. Specifically, we evaluate GOGGLES, Interactive Weak Supervision, Snuba, and a multi-polar variant of Snuba using raw features, PCA, the logits layer of a pretrained ResNet-18, and CLIP logits. We compare all of these to a logistic regression-based few-shot learning baseline, and a Label Propagation-based semi-supervised baseline [44]. In order to robustly assess differences in the types of data domains under which various methods work best, we split our evaluations across image tasks (MNIST, CIFAR-10, Spherical MNIST, Permuted MNIST), diverse tasks (ECG, Navier-Stokes, EMBER), and text tasks (YouTube, Yelp, IMDb). We will go on to describe our analysis and results for determining which AutoWS and feature representation methods work well and do not work well on AutoWS-Bench-101.

### 4.1 Comparison methodology

In addition to presenting a full set of results in Table 1, we adopt the performance profiles methodology of [45] to summarize many of our results in a holistic manner. These are a robust way to visually compare methods in aggregate across noisy evaluations in a large number of environments (i.e., problem settings). The performance profile shows curves where points along each curve represent the fraction of settings where a given method is within a factor of the best method. This factor is denoted $\tau$. Concretely, for each method $s$ in a set of methods $\mathcal{S}$, we plot $\rho_s(\tau) = \frac{1}{|\mathcal{P}|} \left| \left\{ p \in \mathcal{P} : \frac{\text{obj}_{p,s}}{\min_{s \in \mathcal{S}} \text{obj}_{p,s}} \leq \tau \right\} \right|$, where $\mathcal{P}$ is the set of problem settings/environments and $\text{obj}_{p,s}$ is the objective for $s$. The ideal curve has a high fraction, approaching the top-left corner.

The goal of AutoWS is to produce high accuracy and high coverage (that is, to label as many points as possible). When discussing accuracy, we set $\text{obj}_{p,s}$ to be standard classification error. For coverage, we set $\text{obj}_{p,s}$ to be $1 -$ coverage. We set $\mathcal{E}$ to be the set of feature representations, $\mathcal{W}$ to be the set of AutoWS methods, and $\mathcal{D}$ to be the set of datasets that we consider. We evaluate feature representations

$\phi \in \mathcal{E}$ and AutoWS methods $\mathcal{A}_{\mathrm{WS}}^{\phi} \in \mathcal{W}$ by defining their respective sets of environments to be datasets subject to either using a particular AutoWS method, or feature representation, respectively.

## 4.2 Comparison of feature representations and AutoWS methods

**When do foundation models help AutoWS?** First, we study the performance of different feature representation methods across datasets and AutoWS methods. Full results are presented in Table 1. We compare feature representation methods using performance profiles by defining an environment to be a dataset and AutoWS method pair. Specifically, we set $\mathcal{S} = \mathcal{E}$ and $\mathcal{P} = \mathcal{D} \times \mathcal{W}$. We then plot performance profile curves for each feature representation method as shown in Figure 2 (bottom row). We find that CLIP performs quite well on all of the image tasks that we evaluate, whereas ResNet-18 features and raw features were less performant compared to other methods. We postulate that this is because these tasks are somewhat similar to CLIP's training distribution. On diverse tasks, however, CLIP performs much worse—due to its incompatibilities with certain data modalities, such as time series (ECG) and embeddings of tabular data (EMBER), and when using IWS on Navier-Stokes, where the input dimensionality (2) of is too small to generate a large number of LFs. We hypothesize that this is because Navier-Stokes is farther from CLIP's training distribution. Surprisingly, ResNet-18 (which was pretrained on ImageNet) appears to perform quite well on diverse tasks; it achieves the best performance on ECG, and Navier-Stokes under using few-shot baseline and the best performance on EMBER when using Snuba with multipolar LFs. We hypothesize that, while aimed at handling diverse downstream image tasks, ImageNet contains enough semantic information beyond images that it can handle, to at least a rudimentary degree, diverse tasks as well. This observation is in the same vein as prior work which shows that training on (seemingly unrelated) non-linguistic data improves downstream performance on natural language tasks [46]. On text tasks, we find that CLIP underperforms, but we postulate that this is mainly due to the lack of image input in these settings. On the other hand, we find that raw features (i.e., text features extracted using BERT) outperforms other methods. Perhaps unsurprisingly, we find that our few-shot baseline (logistic regression) using raw BERT features outperforms all other methods on text tasks, as this corresponds to standard BERT fine-tuning, which furthermore suggests that this method should be *combined* with AutoWS methods. Our key finding is that **CLIP helps AutoWS on in-distribution image tasks or image-derived tasks. CLIP does not help on diverse out-of-distribution tasks or on tasks where only text input is available, in which raw BERT features help.**

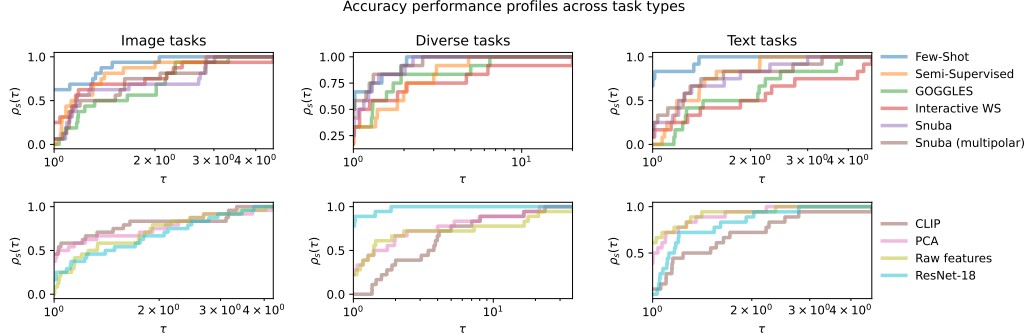

Figure 2: For image tasks (left)—MNIST, CIFAR-10, Spherical MNIST, Permuted MNIST—few-shot learning and CLIP both dominate. For diverse tasks (center)—ECG, Navier-Stokes, Ember—Snuba closes the gap with few-shot learning and CLIP underperforms or is non-applicable. For text tasks (right)—YouTube, Yelp, IMDb—few-shot learning and raw features (i.e., BERT embeddings), while the CLIP text encoder alone underperforms or faces compatibility issues with IWS.

**How does AutoWS compare to few-shot learning?** Next, we compare AutoWS to few-shot learning. Full results are in Table 1. For our performance profiles we set $\mathcal{S} = \mathcal{W}$ and $\mathcal{P} = \mathcal{D} \times \mathcal{E}$. The curves are shown in Figure 2 (top row). We find that few-shot learning typically ourperforms any other method on image tasks, attaining strong average performance. However, we note that in terms of absolute performance, IWS and Snuba perform the best on CIFAR-10 and Spherical MNIST (both when using CLIP), though they are less robust to changes in the underlying feature representation. As

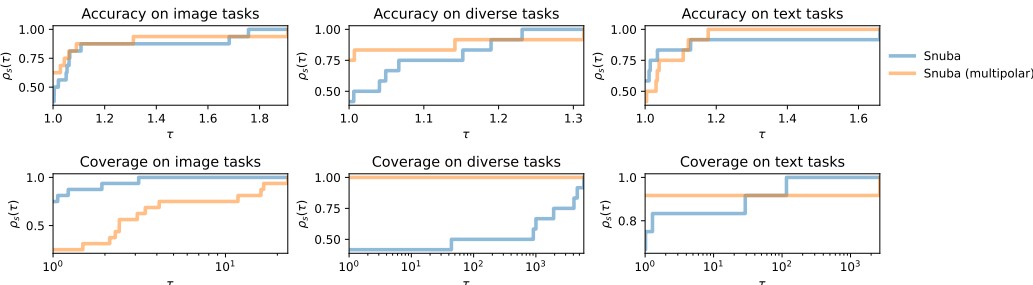

Figure 3: Using multipolar LFs typically improves the performance of Snuba (top). The difference is more pronounced on diverse tasks (top center). Multipolar LFs can result in lower coverage among image tasks (bottom left) which are notably all 10-class classification tasks. Conversely, multipolar LFs result in perfect coverage for the diverse tasks, which have fewer classes (bottom center).

shown in Figure 2 (top center), we find that Snuba (both unipolar and multipolar) perform similarly to few-shot learning on diverse tasks. On Navier-Stokes, multipolar Snuba nearly matches the 100% accuracy of the few-shot baseline and on EMBER, multipolar Snuba outperforms few-shot learning using ResNet-18 features. In terms of absolute performance on all applicable datasets, at least two of the AutoWS methods also outperform raw zero-shot CLIP predictions. We again note that for all text tasks, few-shot learning using raw BERT features (i.e., fine-tuning BERT) outperforms all other methods. Our key finding is that **few-shot learning is a strong baseline with good average performance across tasks, which suggests that it should be combined with AutoWS methods**.

### 4.3 Evaluation of scientific questions using AutoWS-Bench-101

**Are multiclass LFs better than class-specialized LFs?** Next, we directly compare Snuba with unipolar LFs to Snuba with multipolar LFs. To our knowledge, a study of the value of unipolar vs. multipolar LFs has not been previously conducted in the WS literature. The top row of Figure 3 shows that multipolar Snuba can attain slightly better average performance than unipolar Snuba across all feature representations on image tasks, though in terms of absolute numbers in Table 1, unipolar Snuba significantly outperforms multipolar Snuba when using CLIP embeddings (typically the best feature representation for both methods). On diverse tasks, multipolar Snuba outperforms or nearly matches unipolar Snuba, across all feature representations. We furthermore find that on diverse tasks, multipolar Snuba attains perfect coverage in all cases, whereas unipolar Snuba often does not. We note that on these diverse tasks, ECG has 4 classes the other two, Navier-Stokes and EMBER, are binary. We find that on text tasks, multipolar Snuba tends to outperform unipolar Snuba subject to using the best feature representations for each. We also note that multipolar Snuba attains greater than or equal coverage to Snuba on the text tasks (which are all binary) in all but one instance—YouTube with CLIP embeddings. Surprisingly, we find that multipolar Snuba's high coverage does not translate to image tasks, where the number of classes is higher, and unipolar LFs tend to result in much higher coverage on average. We conclude that **multipolar LFs tend to perform better on average but result in lower coverage when the number of classes increases**.

**Do the best methods still cover the label distribution?** A critical aspect for a WS method to be practically useful is that it must be able to produce labels for all of the classes in the label distribution with the same frequency at which they occur in the underlying data distribution. We thus perform an analysis of the coverage of methods in our pipeline subject to using different AutoWS methods and feature representations. We first consider the effect of using different AutoWS methods on coverage. We detail all of the coverage results across IWS, Snuba, and multipolar Snuba in the Appendix (note that GOGGLES by definition always produces 100% coverage). We present performance profile results of our coverage analysis in Figure 4. As before, we find that unipolar Snuba still attains the best coverage across 10-class image tasks, while multipolar Snuba attains the best coverage across diverse tasks and text tasks (with the only exception being YouTube with CLIP embeddings). In the bottom row of Figure 4, we show that methods using CLIP attain, on average, comparatively low coverage across all tasks. On diverse tasks, this can be attributed to the fact that CLIP is only

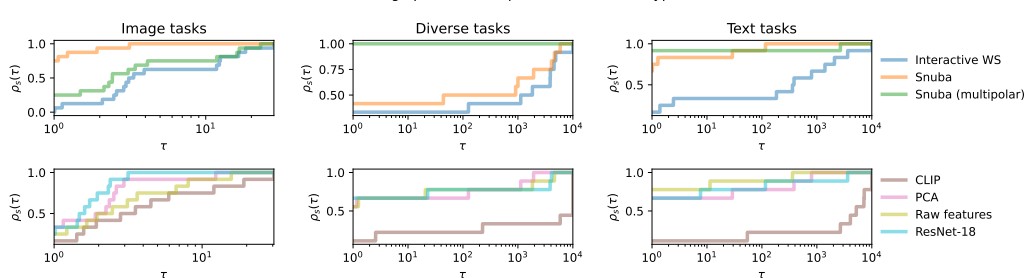

Figure 4: For image tasks (left)—MNIST, CIFAR-10, Spherical MNIST, Permuted MNIST—Snuba attains high coverage, and CLIP usage tends to worsen coverage. For non-image tasks (right)—ECG, Navier-Stokes, Ember—Multipolar Snuba always attains full coverage and CLIP is either non-applicable or attains poor coverage.

applicable to Navier-Stokes, and not to ECG or EMBER (see Figure 4, bottom right). On text tasks, this is partly due to the inapplicability of 2D CLIP logits to IWS (see Appendix for a more detailed explanation of incompatibilities). However, CLIP is applicable to all image tasks, and still attains the worst coverage performance across feature representations (see Figure 4, bottom left). Conversely, ResNet-18 attains the best coverage on image tasks, and coverage is roughly similar between ResNet-18, PCA, and raw features on diverse and text tasks. From this analysis, we conclude that **CLIP usage can hurt coverage**.

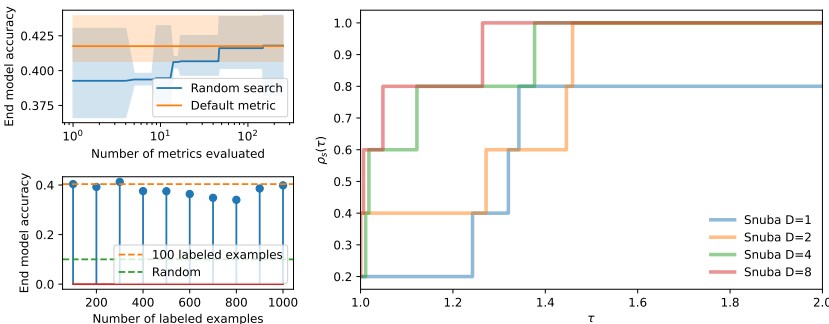

Figure 5: (Top left) Tuning Snuba's LF selection and abstention criteria do not lead to an improvement on MNIST. Results are reported as averages, minima, and maxima over 3 random seeds. We evaluate Snuba on MNIST using PCA and 500 feature combination samples with cardinality $D = 2$. (Bottom left) Increasing the number of labeled examples does not improve the performance of Snuba. (Right) Increasing cardinality can lead to some improvement with Snuba.

## 4.4 Ablation study of Snuba

We conclude our analyses with an ablation study of Snuba, the best performing AutoWS method that we evaluated. We consider the effects of varying the cardinality (the number of features that each LF can consider), the LF selection criterion used by Snuba (F1 score), and the effect of increasing the initial number of labels beyond the original 100.

**Does increasing cardinality help Snuba?** Snuba's cardinality parameter dictates how many components of the feature representation each LF can consider during its LF synthesis procedure. We evaluate the aggregate effect on performance of varying this parameter using performance profiles, as shown in Figure 5 (left). In this case, we set $\mathcal{S}$ to be Snuba with cardinalities $D = 1, 2, 4,$ and 8. We consider only CLIP embeddings and all datasets on which CLIP is applicable: MNIST, CIFAR-10, Spherical MNIST, Permuted MNIST, and Navier-Stokes. This limits the maximum cardinality that we can consider to 10, which allows us to explore the full space of possible improvements for Snuba along the axis of cardinality. In these settings, increasing cardinality improves performance up to a

point. Beyond $D = 4$, we observe that Snuba ceases to improve much. From this, we conclude that **increasing cardinality can lead to performance improvements up to a point**.

**Is F1 the right metric for MNIST when using Snuba?** Given the poor performance of Snuba on MNIST, we evaluate Snuba's default choice to use F1 scores for LF selection and for determining abstain thresholds—micro F1 and weighted F1, respectively. We do so by first defining a search space over all applicable classification metrics available in scikit-learn [47]. These include Snuba's default choices, accuracy, balanced accuracy, precision, recall, Cohen's kappa coefficient, Jaccard index, Matthews correlation coefficient, and using the micro-averaged F1 score for both LF selection and to determine abstain thresholds. We perform a random search over convex combinations of these 9 metrics by sampling configurations from a Dirichlet distribution with parameter $\alpha = \mathbf{1}_9$. As shown in Figure 5, we find no appreciable improvement over the default configuration. However, the best configuration found by random search has majority weight on micro-averaged F1 score, which only slightly differs from the default configuration in Snuba. These results indicate that **Snuba's poor performance on MNIST is not trivially due to a poor choice of metric for LF selection**.

**Do more labels help Snuba?** We ask if we can further improve performance on MNIST by increasing the number of labeled examples. In this experiment, we varied the number of labeled examples given to Snuba from 100 to 1000. We set the cardinality to $D = 2$ and use raw pixel features. For efficiency, we randomly subsample 1000 feature combinations so as to avoid exploring the entire space of pairwise feature combinations. We answer the question in the negative: our results in Figure 5 show that increasing the number of labeled examples does not improve the performance of Snuba. This suggests that **any performance with Snuba in this setting is likely algorithmic, and not due to a lack of labeled examples**.

## 5    Conclusion

In this work, we proposed AutoWS-Bench-101, a benchmarking framework for AutoWS methods and beyond on a diverse set of classification tasks. We found that foundation models can improve the performance of AutoWS methods on in-distribution tasks but they can also hurt coverage, sometimes dramatically, on out-of-distribution tasks. We furthermore found that AutoWS methods using multipolar LFs might outperform unipolar LFs on average, but can result in lower coverage as the number of classes increases. One limitation is that we only considered CLIP—but there are many other applicable methods to be studied in future work. We hope this work takes a first step towards understanding how to build best-of-all-worlds automated weak supervision.

## Acknowledgments

We are grateful for the support of the NSF (CCF2106707), the American Family Funding Initiative and the Wisconsin Alumni Research Foundation (WARF). Any opinions, findings and conclusions or recommendations expressed in this material are those of the author(s) and do not necessarily reflect the views of any of these funding agencies.

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
