# OpenReview forum: "AutoWS-Bench-101: Benchmarking Automated Weak Supervision with 100 Labels"
_NeurIPS.cc/2022/Track/Datasets_and_Benchmarks — NeurIPS 2022 Datasets and Benchmarks _

### Official Review · Reviewer_mCjw · 2022-07-18
**Benchmark for weak supervision - could be more comprehensive**

**Rating:** 5
**Confidence:** 3

**Strengths:**

1. The paper adds an additional analysis aspect of different features for weak supervision that was previously unexplored.
2. The comparison using and findings regarding the use of foundation models for weak supervision are quite relevant to subsequent research on foundation models.

**Weaknesses:**

1. The benchmark is significantly smaller in number of datasets and methods compared to WRENCH, an existing benchmark for Auto-WS evaluation.
2. Not all methods used were analyzed further. For instance the paper focused much more on the analysis of the method Snuba.
3. Paper has problems with clarity (see below).
4. Other than the paper's contributions, it is unclear why one may use this benchmark for for evaluating an Auto-WS method over WRENCH. Ln 80 saying "understanding mechanics" is somewhat vague.
5. Not a deal-breaker, but the paper can benefit from additionally using semi-supervised learning methods as a baseline besides simply using few-shot learning.

Most questions that were unclear from the paper have been mentioned under "Clarity" below.

**Additional Feedback:**

All covered above

**Clarity:**

The paper is not very easy to navigate. Following are some questions that were unclear :
1. It seems CLIP embeddings were not used for ECG and EMBER datasets. How can CLIP be compared with other methods in Fig 2 on the right?
2. Ln 116 : What was Resnet-18 pre-trained on?
3. Ln 221 mentions CLIP features help on in-distribution tasks, but they help on spherical and permuted MNIST as well, which are not "in-distribution"/natural images. Maybe the conclusion from this experiment needs to be more nuanced.
4. Could not find details regarding how the few-shot baseline abstained.
5. Table 5 : Unclear why the Navier-Stokes class descriptions are so different.

Other comments:
1. Table 1 would be much clearer with a mention of what the metrics are. It is easy to think the metric is error on looking at Table 1 after Sec 4.1.
2. Brief descriptions of methods evaluated, including brief descriptions of labeling and abstention, can add a lot to the paper's readability

- One typo that I spotted : ln 172 : remove "with"

**Correctness:**

Most evaluations seem correct apart from from some minor questions regarding CLIP in Fig 2 (see below).


**Documentation:**

Dataset seems adequately documented.

**Relation To Prior Work:**

Not quite (see pt 4 under weaknesses above). Also, the paper could benefit from a discussion of how few-shot learning or semi-supervised learning compares/contrasts to Auto-WS methods.


**Summary And Contributions:**

The paper introduced a benchmark for evaluation of automated weak-labeling methods called AutoWS-Bench-101. The evaluation included multiple methods and features used for data, on multiple image and non-image classification tasks. Compared to a prior benchmark on the same (WRENCH) however, the current benchmark is significantly smaller, however using different features as an additional aspect of analysis.

---

> ### Author Response · Authors · 2022-08-17
> **Response to Reviewer mCjw (Part 1/2)**
>
> We thank the reviewer for their review and for praising our analyses of AutoWS using various features and for the relevance of studying foundation models. We have provided clarifications below.
>
> > “The benchmark is significantly smaller in number of datasets and methods compared to WRENCH, an existing benchmark for Auto-WS evaluation.”
>
> The key difference here is that WRENCH is not a benchmark for AutoWS evaluation—it evaluates a different problem from the one we investigate in our paper. Our paper evaluates methods that **generate LFs** (i.e. AutoWS methods), particularly on data types where obtaining LFs is difficult. In contrast, WRENCH evaluates **WS label models** where the LFs are assumed to be fixed. To highlight this difference, the WS methods in WRENCH are not directly applicable to our benchmark, as they require existing LFs. This crucial distinction highlights the need for our benchmark. Indeed, these are complementary benchmarks that together cover both aspects of weak supervision: producing labeling functions (AutoWS-Bench-101) then selecting and training a label model (WRENCH). To your former point, we have included three additional text datasets and an additional label propagation baseline (see the general response for more details).
>
> > “Not all methods used were analyzed further …”
>
> We thank the reviewer for raising this important point. We have **included additional ablation studies of the other AutoWS methods.** We now provide an ablation study of GOGGLES in Appendix C, and ablation study of Interactive WS in Appendix D (please refer to the general response for more details).
>
> > “Other than the paper's contributions, it is unclear why one may use this benchmark for for evaluating an Auto-WS method over WRENCH …”
>
> An AutoWS method can indeed be applied to the datasets in WRENCH. However, the main motivation for using AutoWS over WS is to enable the usage of weak supervision in settings where obtaining LFs is difficult—this is not the case for WRENCH, nor is it the motivation for their benchmark. Our benchmark evaluates AutoWS methods (as opposed to WS methods) on datasets that are more difficult for AutoWS than the ones found in WRENCH.
>
> > “... the paper can benefit from additionally using semi-supervised learning methods as a baseline besides simply using few-shot learning.”
>
> We thank the reviewer for this important suggestion. We have included a semi-supervised baseline: label propagation. Please refer to the general response for additional details.

---

> > ### Author Response · Authors · 2022-08-17
> > **Response to Reviewer mCjw (Part 2/2)**
> >
> > > “... Following are some questions that were unclear :
> > > 1. It seems CLIP embeddings were not used for ECG and EMBER datasets. How can CLIP be compared with other methods in Fig 2 on the right?
> > > 2. Ln 116 : What was Resnet-18 pre-trained on?
> > > 3. Ln 221 mentions CLIP features help on in-distribution tasks, but they help on spherical and permuted MNIST as well, which are not "in-distribution"/natural images. Maybe the conclusion from this experiment needs to be more nuanced.
> > > 4. Could not find details regarding how the few-shot baseline abstained.
> > > 5. Table 5 : Unclear why the Navier-Stokes class descriptions are so different.”
> >
> > We provide clarifications to each of these points below.
> > 1. We have added a clarification in section 4.2 that CLIP is incompatible with ECG and Ember due to the fact that both datasets comprise only feature vectors, and that this is partly why CLIP underperforms in these settings.
> > 2. ResNet-18 was pretrained on ImageNet.
> > 3. Thank you for pointing this out. We have clarified this conclusion to include image-derived tasks.
> > 4. Our few-shot and semi-supervised baselines have no abstention mechanism by definition – they always have 100% coverage.
> > 5. The Navier-Stokes class descriptions are based on the viscosity parameter for the data in class 0 vs. class 1. This viscosity parameter has a physical interpretation that corresponds to turbulence – a low viscosity corresponds to a turbulent flow regime, while a high viscosity does not. This description corresponds exactly to our labels for this dataset: binary labels for turbulent flow classification.
> >
> > > “Table 1 would be much clearer with a mention of what the metrics are …”
> >
> > Thank you for pointing this out—we have updated the caption of Table 1 accordingly. The metric is test set accuracy.
> >
> > > “Brief descriptions of methods evaluated, including brief descriptions of labeling and abstention, can add a lot to the paper's readability”
> >
> > We describe the AutoWS methods that we consider: Snuba, Interactive Weak Supervision, and GOGGLES in Section 3.2. We have added details to this section regarding how each of these methods abstain. To summarize, Snuba infers a per-LF confidence score threshold that is used for abstention. We rely on Snuba’s LF synthesis procedure, including abstentions, in Interactive Weak Supervision. Lastly, we clarify that GOGGLES does not abstain.
> >
> > > “Not quite (see pt 4 under weaknesses above). Also, the paper could benefit from a discussion of how few-shot learning or semi-supervised learning compares/contrasts to Auto-WS methods.”
> >
> > In Section 1, we describe how AutoWS frameworks resemble few-shot learners, but are **omnivorous frameworks,** meaning that they can incorporate signals from multiple sources including other few-shot, zero-shot, or semi-supervised learners. We clarify that the same is not generically true of few-shot, zero-shot, or semi-supervised learning frameworks on their own.

---

> > > ### Comment · Reviewer_mCjw · 2022-08-19
> > > **Thanks for the response.**
> > >
> > > Thanks for the response. Most of my concerns have been addressed, however the work still seems very incremental over WRENCH in the evaluation of Auto-WS vs WS. There does not seem to be a need for a separate benchmark for Auto-WS when the main methods compared are Snuba and GOGGLES. The authors of GOGGLES have compared their method to Snuba in their paper. I maintain my initial rating.
> > >
> > > **Follow up**:
> > > Re incompatibility of CLIP : in the equation for $\rho$, what is $obj_{p, s}$ for $p =$ ECG or Ember and $s =$ CLIP (since these would be required for computing $\rho$ if I'm not wrong?)

---

> > > > ### Author Response · Authors · 2022-08-21
> > > > **The goal of our paper**
> > > >
> > > > We thank the reviewer for the reply. However, we respectfully disagree with this assessment. We would like to clarify that **the goal of our paper is to understand what makes AutoWS techniques work at a fundamental level—this has not been the goal of prior work.** We believe these techniques have the potential, when used in concert with foundation models, to achieve excellent performance across diverse problem settings with little human intervention.
> > > >
> > > > This is exciting, but to deliver on this promise, we need to understand the basic mechanics of such techniques. Our work does so by:
> > > > - **breaking down existing techniques into modular components,** and introducing new ones (e.g., multiclass Snuba variants),
> > > > - introducing an overall workflow that **enables mixing-and-matching the modular components,**
> > > > - **measuring multiple metrics against diverse tasks** (including text, image, tabular, and other problems).
> > > >
> > > > Indeed, this goes far beyond the existing works.
> > > >
> > > > **Re: follow up:** As for incompatible methods/feature extractors/datasets, we set the default classification error to be 1.0 in order to compute performance profile curves.

---

### Official Review · Reviewer_s6Jy · 2022-07-21
**A useful benchmark for automatic weak label generation**

**Rating:** 7
**Confidence:** 4
**Correctness:** Yes

**Strengths:**

1. The paper studies a very interesting topic of automatic weak label generation. To the best of the reviewer's knowledge, this research topic has not been widely explored in previous studies. It will be great if indeed this work will create a unified framework to standardize AutoWS evaluation.
2. The paper provides extensive experiments for evaluating different methods with detailed discussions on the performance of different AutoWS methods.
3. The writing is overall clear and easy to follow.
4. The code has been released with documentation. As it is adapted from the Wrench benchmark, it will be easy for WS researchers to follow this new benchmark.

**Weaknesses:**

1. For the background part, I think it would be better to give a detailed description of the AutoWS setting, as it is very different from the traditional WS task (in Wrench paper). I recommend the author use some mathematical notations to define the problem formally.
2. It would be better to extend the tasks to other domains such as NLP.  Is it possible to run some of the baselines by using the datasets in the Wrench benchmark?
3. The few-shot baselines seem to be very weak (using logistic regression only), it would be better to use more advanced techniques.
4. The analysis for the snuba method is interesting. Is it possible to give more analysis for other baselines?


**Additional Feedback:**

In table 1, it remains unclear how to use ResNet-18 for ECG and Ember since they are not standard image classification tasks. Could the author elaborate more on this?

**Clarity:**

Yes, most of the parts are clear and easy to follow. One recommendation is that the authors could try to use more mathematical notations/equations to better describe the problem setup and baseline methods.

**Documentation:**

The Documentation looks good to me.

**Relation To Prior Work:**

Yes, this paper introduces the related work in detail and covers most of the recent works.
It would be even better to discuss some autoWS methods for NLP (listed below):
- Adaptive rule discovery for labeling text data (SIGMOD 2021)
- Prompt-based rule discovery and boosting for interactive weakly-supervised learning (ACL 2022)
- Nemo: Guiding and contextualizing weak supervision for interactive data programming. (arXiv:2203.01382)
- Automatic Rule Induction for Efficient Semi-Supervised Learning (arxiv 2205.09067)

**Summary And Contributions:**

This paper introduces a novel benchmark, AutoWS-Bench-101 for evaluating different AutoWS methods. Different from the previous WS benchmark WRENCH which evaluates different label models and end models with some heuristics as the labeling function, this benchmark aims to generate weak supervision sources directly from the feature space. This benchmark contains massive experiments with zero-shot predictions from foundation models, and evaluates with multiple AutoWS methods, including very recent approaches. The codebase and datasets have been publicly released.

---

> ### Author Response · Authors · 2022-08-17
> **Response to Reviewer s6Jy**
>
> We thank the reviewer for their thoughtful and thorough review, as well as for their praise of the novelty of our benchmark, our experimental results, the clarity of the writing, and of our code. We have responded to each of the reviewer’s points below.
>
> > "... I recommend the author use some mathematical notations to define the problem formally."
>
> This is a great suggestion! We have added this to section 2.
>
> > "It would be better to extend the tasks to other domains such as NLP. Is it possible to run some of the baselines by using the datasets in the Wrench benchmark?"
>
> We thank the reviewer for raising this excellent point – we agree with this. We have added three text datasets to AutoWS-Bench-101: YouTube, Yelp, and IMDb (each of which are additionally used in WRENCH). See the general response for more details.
>
> > "The few-shot baselines seem to be very weak (using logistic regression only), it would be better to use more advanced techniques."
>
> We have included an additional semi-supervised baseline: the **classical label propagation technique** (please refer to the general response for more details). We plan on adding further  baselines to our evaluation.
>
> > "The analysis for the snuba method is interesting. Is it possible to give more analysis for other baselines?"
>
> We thank the reviewer for raising this point and for the praise of our Snuba analysis. We have included additional analyses GOGGLES and Interactive WS in the Appendix (Appendices C and D, respectively, and see the general response for a more detailed overview).
>
> > "It would be even better to discuss some autoWS methods for NLP …"
>
> Thank you for these references! We have added some commentary about these methods to section 2.
>
> > "In table 1, it remains unclear how to use ResNet-18 for ECG and Ember since they are not standard image classification tasks …"
>
> Yes! See the general response.

---

### Official Review · Reviewer_kYXS · 2022-07-22
**Interesting benchmark**

**Rating:** 7
**Confidence:** 2
**Correctness:** Correct as far as I can tell.
**Clarity:** Yes.

**Strengths:**

+ interesting problem
+ well motivated

**Weaknesses:**

- conclusions seem to be based on essentially single data point

**Additional Feedback:**

The paper is well-written and motivates the need for this benchmark set. The
main area where it falls short is the empirical evaluation, which seems to be
based on a single foundation model, which makes the conclusions that the authors
draw with respect to the goals of their evaluation somewhat tenuous.

That said, the proposed dataset is interesting and should be useful to
researchers in practice.

The caption for Table 1 should explain what the numbers mean.

Update after rebuttal:

I believe that my main point has been addressed with the extended evaluation and am happy to recommend acceptance now.

**Documentation:**

Sufficient.

**Relation To Prior Work:**

Yes.

**Summary And Contributions:**

The paper proposes a benchmark for automated weak supervision that designs
labeling functions. The authors describe the problem and data and present
empirical results.

---

> ### Author Response · Authors · 2022-08-17
> **Response to Reviewer kYXS**
>
> We thank the reviewer for their thoughtful review and for the praise of the motivation of our work. We have responded to each of your comments below.
>
> > “conclusions seem to be based on essentially single data point”
> …  “... which seems to be based on a single foundation model, which makes the conclusions that the authors draw with respect to the goals of their evaluation somewhat tenuous.”
>
> We thank the reviewer for raising this important point. We agree that including additional foundation models would improve our evaluation. One challenge that we encounter with selecting foundation models for our datasets is that many do not operate on the modalities used in our benchmark. CLIP itself isn’t applicable to two of our tasks (ECG and Ember) because these datasets consist of only feature vectors (as opposed to text or image-like tensors). To address this challenge, we have included three new text datasets **complete with results which use BERT features.**
>
> > “The caption for Table 1 should explain what the numbers mean.”
>
> The caption has been updated. The numbers in Table 1 are test set accuracy scores.

---

> > ### Comment · Reviewer_kYXS · 2022-08-18
> > **Thank you.**
> >
> > Thank you for your response!

---

### Official Review · Reviewer_dFcd · 2022-07-25
**In general, it is a good dataset paper. But the authors are suggested to add more datasets.**

**Rating:** 6
**Confidence:** 3
**Correctness:** The evaluation methods and experiment…

**Strengths:**

1. This paper proposed a benchmarking framework for an important research line of methods --- AutoWS.
2. This paper provided the sufficient details e.g., the SOTA baseline experimental results and all experiment codes.


**Weaknesses:**

1. This AutoWS-Bench-101 benchmark contains several image datasets but only one dataset in other domains. There are many other time-series and tabular datasets. The conclusion for some field based on the experimental results of only one dataset is not so convincing. The authors are suggested to add more datasets in each domain except for image classification.

**Additional Feedback:**

The authors are also encouraged to add NLP datasets in this benchmark. But it’s minor.

**Clarity:**

This paper is generally well-written. I only have several minor comments. Several examples are as follows:
1. Table 1 and one of the descriptions about Table 1 in the main body are suggested to be in the same page. Table 1 is in page 5 now and the text in page 5 doesn’t have the description about Table 1.
2. There are many “N/A” in Table 1 & 2. The authors are suggested to provide some explanations about them in the captions or footnotes.
3. In line 105-106, the authors said “…We are especially interested in settings far from the ambit of conventional weak supervision---most often used in text applications…”. From my experience, weak supervision is also often used in Computer Vision and other applications. We can discuss the correctness of this claim.


**Documentation:**

This paper provided the sufficient details on documentation.

**Ethics:**

The authors discussed the ethics issues in the supplementary materials.

**Relation To Prior Work:**

This paper clearly described the relations to prior work.

**Summary And Contributions:**

This paper proposed a benchmarking framework for AutoWS --- AutoWS-Bench-101 on a diverse field of tasks. They provided the baseline experimental results of different AutoWS methods on seven datasets. They also conducted the experiments utilizing different foundation models and multipolar LFs to observe the pros and cons.

---

> ### Author Response · Authors · 2022-08-17
> **Response to Reviewer dFcd**
>
> We thank the reviewer for the thoughtful review! We have responded to each of your comments below.
>
> > "This AutoWS-Bench-101 benchmark contains several image datasets but only one dataset in other domains …"
>
> Among our original seven datasets, only four of them are image datasets (MNIST, CIFAR-10, Spherical MNIST, and Permuted MNIST). ECG is a non-image time-series dataset, Navier-Stokes comprises partial differential equation data, and EMBER consists of tabular features. Moreover, although Spherical MNIST and Permuted MNIST are indeed image-derived, these are not natural images. Additionally, **we have added three text datasets** to AutoWS-Bench-101: YouTube, Yelp, and IMDb. Please refer to the general response for more details.
>
> > "Table 1 and one of the descriptions about Table 1 in the main body are suggested to be in the same page …"
>
> We have attempted to move tables and figures closer to where they are first mentioned.
>
> > "There are many “N/A” in Table 1 & 2 …"
>
> We have added clarifications about these in Appendix G, and we refer to the Appendix in the main text. To summarize Appendix G, CLIP does not support the vector input comprising the data in ECG or Ember, respectively. Additionally, CLIP features (in the form of logits) are not supported by Interactive WS on binary classification tasks, as the resulting 2D logit does not contain enough entries to produce the minimum number of LFs required by Interactive WS using the LF generation rule we use throughout the paper—the synthesis procedure from Snuba.
>
> > "... weak supervision is also often used in Computer Vision and other applications. We can discuss the correctness of this claim."
>
> It is indeed the case that weak supervision has been applied to Computer Vision and other applications in the past. The context is roughly the following: WS requires primitives to construct LFs, and since text is so amenable to this, text is the most common application of WS. Primitives are more difficult to obtain in image settings. The number of works that represent each of these domains reflects this; many WS works mainly involve text applications, some involve image applications, and few works move beyond these two domains, as primitives are often even more difficult to obtain.
>
> > "The authors are also encouraged to add NLP datasets in this benchmark. But it’s minor."
>
> We thank the reviewer for raising this excellent point—we agree. **We have added three text datasets** to AutoWS-Bench-101: YouTube, Yelp, and IMDb. See the general response for more details.

---

### Official Review · Reviewer_WCRn · 2022-07-26
**AutoWS-Bench-101: benchmark for automated weak supervision**

**Rating:** 7
**Confidence:** 4

**Strengths:**

Considering the potential impact of automated weak supervision in many domains, especially when using "foundation models" as labeling function co-operating w/ other sources, I find the authors' contribution, i.e. a benchmark for such techniques, very relevant for the research community.  The benchmark allows to easily perform experiments to better characterize these techniques. The codebase is well structure providing all the necessary abstractions to quickly set up an experiment environment and easily allowing its extension for future work and research directions.

**Weaknesses:**

line 105-107: while explicitly explained by the authors, I still think that the benchmark should include one or more datasets for text applications. This is because "foundation models" are extremely popular in NLP (in the form of large pre-trained language models) and allowing to investigate their impact in AutoWS setting would make the benchmark more "complete".  With this being said, if the authors find that this is already explored elsewhere, (a) pointer(s) for the interested reader should be provided.


**Additional Feedback:**


- line 164-165, and 168-169: for IWS "we use Snuba’s LF synthesis procedure to generate a large set of LFs" but then "we run the Interactive WS LF selection procedure once per class to produce unipolar LFs". I am not familiar w/ this frameworks, but if Snuba's can be adapted to be multi-class, why not use the multi-class version for IWS?

- line 218-219: I must admit I am confused about this, but this may be due to my lack of a thorough background in this domain.
What is the motivation behind using features from an image model like ResNet for non-image datasets? Is this "common" in the (auto)WS setting? Does this make sense at all? What is really telling us the fact that a logistic regression model on top of ResNet logits (a model trained on natural images!) can achieve an accuracy of 0.47 on an electrocardiogram dataset?

- line 260-262: maybe I am missing something but why is this the case (setting aside the previous point)? Results are reported on ECG and EMBER w/ ResNet features, and from my understanding CLIP is still an image model like ResNet, simply needing a natural lanugage prompt.




**Clarity:**

The paper is very well written and I could follow all the arguments (almost) without effort.

There are some minor adjustments that in my opinion would make it even better:
- Table 1: Caption should explicitly state what is being measured (what is performance in this case). Which split (test/dev)?
- line 150: the few-shot learning baseline is introduced almost incidentally in section 3.2 "implementation details".
Arrived at section 4 "Analysis", a reader like me either did not notice it or has already forgotten this detail.  For clarity this should be repeated in 4, or maybe in the caption for Table 1.
- line 269: cardinality is defined here but is mentioned before as well in line 166.


**Correctness:**

As far as I understand it the evaluation methods and the experiments performed are appropriate to the type of research questions investigated.

**Documentation:**

The repository where the benchmark code is hosted provides all the details to reproduce the work.

**Ethics:**

In my opinion the benchmark does not present any ethical concern.

**Relation To Prior Work:**

The prior work is cited correctly and the authors clearly define the contributions their work present.

**Summary And Contributions:**

Automated Weak Supervision (AutoWS) methods aim to bring a step further WS methods by providing automated approaches to create Labeling Functions (LF) which are then used to build labeled datasets for training supervised models.
The authors introduce AutoWS-Bench-101, a benchmark for automated weak supervision methods. The benchmark aims to be a framework to allow comparison among multiple AutoWS techniques and investigate the effect of its components such as the representations used in automated LFs.
The benchmark offers multiple implementations of AutoWS methods s.a. Snuba and GOGGLES and a wide selection of a datasets to experiment with, mostly in the image domain.
With this benchmark the authors then perform a series of experiments to better characterize AutoWS methods. These include for instance assessing the impact of using foundation models (pre-trained models such as ResNet) as labeling functions and how AutoWS compare to (a) few-shot learners (which can be foundation models themselves) (b) hand-designed LFs.

---

> ### Author Response · Authors · 2022-08-17
> **Response to Reviewer WCRn**
>
> We thank the reviewer for the thoughtful and thorough review as well as for the comments about the relevance of our work along with positive comments about our codebase. We have responded to each of the points below.
>
> > "... I still think that the benchmark should include one or more datasets for text applications …"
>
> Thank you for raising this excellent point–we agree with this. We have **added three text datasets** to AutoWS-Bench-101: YouTube, Yelp, and IMDb. See the general response for more details.
>
> > "Table 1: Caption should explicitly state what is being measured…"
>
> The scores presented in Table 1 are test set accuracy scores. We’ve updated the Table 1 caption to reflect this.
>
> > "line 150: the few-shot learning baseline is introduced almost incidentally …"
>
> We have updated this in Section 4 and we have updated the table to make this more clear.
>
> > "line 269: cardinality is defined here but is mentioned before as well in line 166."
>
> Thank you for catching this! We have updated the paper to define cardinality earlier.
>
> > "... if Snuba's can be adapted to be multi-class, why not use the multi-class version for IWS?"
>
> We found that the open source implementations of Snuba and IWS were both only defined for unipolar LFs, so we initially followed a one vs. rest scheme when using both of these in multiclass settings. Indeed, we adapted Snuba to use multipolar LFs, but found that this was a non-trivial adaptation. We are working on doing so for IWS as well.
>
> > "... What is the motivation behind using features from an image model like ResNet for non-image datasets? …"
>
> As far as we are aware, this is not common nor has a pretrained ResNet been previously used with AutoWS on diverse tasks. We simply chose to use this architecture because it type-checks for a lot of data types in the sense that the architecture itself uses global average pooling and can handle input such as single-channel feature vectors as opposed to image tensors alone. We were surprised to find, however, that this **actually led to strong performance on diverse non-image tasks!** We provide additional commentary about this in the general response.
>
> > "line 260-262: maybe I am missing something but why is this the case (setting aside the previous point)?"
>
> Thank you for pointing this out. While CLIP is indeed a joint image and text model, we found that it did not support feature vectors in the same way that the ImageNet variant of the ResNet architecture does. We plan to work on alternatives that will enable this comparison.

---

> > ### Comment · Reviewer_WCRn · 2022-08-23
> > **Response to authors**
> >
> > Thank you for addressing my comments.
> >
> > > What is surprising—and remarkable—is how well a ResNet-18 model, pretrained on ImageNet, does on our diverse tasks (ECG, Navier-Stokes, and Ember, see Figure 2, center bottom). This is one of the most interesting findings in our work.
> >
> > This is indeed fascinating. It actually reminded me of [this paper](https://aclanthology.org/2020.emnlp-main.554/), where they find that (and I quote) "training on non-linguistic data [...] improves test performance on natural language, despite no overlap in surface form or vocabulary".  Maybe it makes sense to cite it?
> >
> > Final very small suggestion: I was looking again at Table 1 and there is a lot of information there, how about underlining the scores of the second best method? This way the reader can make comparisons more quickly.

---

> > > ### Author Response · Authors · 2022-08-23
> > > **Thank you for the suggestions!**
> > >
> > > Thank you for these suggestions!
> > >
> > > This paper indeed seems related to the phenomenon that we observed with ResNet-18 features---we have cited the paper accordingly. We have also underlined the second best methods in Table 1 for readability---thank you for suggesting this!

---

### Official Review · Reviewer_r9PR · 2022-07-27
**New benchmarks for auto weak supervision**

**Rating:** 7
**Confidence:** 3
**Correctness:** The claims seem correct
**Clarity:** The paper is largely written well.

**Strengths:**

This paper has many strengths:
* There is a need for this study in the community and future works may find value in the experimental findings
* The experiments span a range of settings and findings can likely inspire future studies and empirical guidelines
* The experiments cover multiple data modalities

**Weaknesses:**

This paper could be improved with the following considerations:
* The motivation flip-flops a little bit: sometimes the authors say LFs are super cheap and easy (Line 27 and 72, for example), other times they are expensive and hard (Line 37 for example). I recommend sticking to a clearer story or qualifying these statements a little more.
* The experimental results section could be strengthened by further interpretation of why different methods may be performing better or worse in different cases.

**Additional Feedback:**

In the experiments section, I suggest putting tables and figures on the same pages where they are mentioned. Otherwise, it's hard to flip back and forth to keep the points clear.

**Documentation:**

The github repo looks well-documented and reproducible (though I did not try)

**Ethics:**

No ethical concerns

**Relation To Prior Work:**

Relation to prior work is clear

**Summary And Contributions:**

This work benchmarks recent automated weak supervision methods on seven datasets spanning multiple modalities. The authors present a head-to-head comparison of AutoWS methods with special motivation from the rise of foundation models---the relationship between pretrained models and AutoWS is only just beginning to be explored. The results indicate that foundation models unsurprisingly may not be helpful on out-of-distribution data.

---

> ### Author Response · Authors · 2022-08-17
> **Response to Reviewer r9PR**
>
> We thank the reviewer for the thoughtful review as well as for the praise of our experimental findings and coverage of multiple data modalities. We have responded to each of your points below.
>
> > "The motivation flip-flops a little bit …"
>
> Thank you for raising this point—this distinction is precisely **the distinction between WS and AutoWS!** In WS, LFs are assumed to be cheap and easy to obtain. This is often the case for text tasks, but it is more difficult to write LFs for richer modalities such as images or much more complex modalities such as PDEs. It is in these situations that AutoWS shows promise, and this is exactly why we’ve designed our benchmark around such tasks.
>
> > "The experimental results section could be strengthened by further interpretation …"
>
> We thank the reviewer for this suggestion. We have added **additional explanations and content to the experiments section** to address this. In particular, we have added some commentary regarding our perhaps most surprising result: ResNet-18 pretrained on ImageNet appears to outperform other feature representations on diverse tasks beyond images (ECG, Navier-Stokes, and Ember). We have provided additional details about this in the general response.
>
> > "In the experiments section, I suggest putting tables and figures on the same pages …"
>
> We have attempted to move tables and figures closer to where they are first mentioned.

---

> > ### Comment · Reviewer_r9PR · 2022-08-18
> > **Response to authors**
> >
> > Thank you for your response, I have raised my score accordingly. For the motivation flip-flopping, I suggest making sure it's obvious that ease-of-labeling depends on modalities and settings as you work through the introduction because this is a strong, convincing point!

---

### Author Response · Authors · 2022-08-17
**General Response**

We thank the reviewers for their thoughtful comments and their excellent questions. These have led to improvements in both the clarity of our paper and the ground that we cover in our experimental results. We have organized a general response to common questions and requests for additional experimental results, detailed below. Afterwards, we respond to each reviewer in-depth.

1. **New Datasets:** We have added **three NLP datasets** to our benchmark: YouTube, Yelp, and IMDb. The results for these can be found in Table 1 and throughout section 4. We have separated the analyses into three categories by task type: image tasks, diverse tasks, and text tasks (whereas before, it was image tasks and non-image tasks).
2. **New Baselines:** We have added a **classic semi-supervised baseline, label propagation.** The results can be found in Table 1 and Figure 2.
3. **Additional Ablation Studies:** We have conducted additional ablation studies for GOGGLES and IWS—these can be found in Appendix C and D, respectively.
4. **On the Surprising Performance of an ImageNet-trained ResNet-18 on ECG, Navier-Stokes, and Ember:** Due to the use of global average pooling in its architecture, the ImageNet variant of ResNet-18 can handle a variety of input shapes, including 1D feature vectors. What is surprising—and remarkable—is how well a ResNet-18 model, pretrained on ImageNet, does on our diverse tasks (ECG, Navier-Stokes, and Ember, see Figure 2, center bottom). This is one of the most interesting findings in our work. We hypothesize that, while aimed at handling diverse downstream image tasks, ImageNet contains enough semantic information beyond the purely visual that it can handle, to at least a rudimentary degree, some non-image tasks as well. We have added commentary about this in Section 4.2. Part of our ongoing work is to establish how far this result can be taken.

---

### Meta-Review · Area_Chair_rT8f · 2022-09-11

**Recommendation:** Accept
**Confidence:** 4

**Metareview:**

All reviewers except one recommend acceptance. The reviewer recommending rejecting the paper (score 5) raises valid concerns, e.g., the lack of semi-supervised baselines and the comparison to a prior benchmark for weak supervision (WRENCH). The authors addressed some of the concerns (e.g., by adding a semi-supervised learning baseline). Regarding the comparison to WRENCH: It is true that WRENCH is similar, but the exact problem benchmarked by WRENCH and AutoWS-Bench-101 is different: in the former, the labeling functions are assumed to be given, in the latter, the labeling functions are learned. I find this difference sufficient for a separate benchmark.

Overall I recommend accepting the paper. I strongly encourage the authors to take all the reviewer comments into account, especially the comparison to baselines form semi-supervised learning. As I understand, automatic weak supervision fits the same abstract problem statement as semi-supervised learning. Hence a comprehensive comparison to semi-supervised learning is essential.

---

### Decision · Program_Chairs · 2022-09-16

Accept